# Metabolic Engineering of Microorganisms to Produce Pyruvate and Derived Compounds

**DOI:** 10.3390/molecules28031418

**Published:** 2023-02-02

**Authors:** Qian Luo, Nana Ding, Yunfeng Liu, Hailing Zhang, Yu Fang, Lianghong Yin

**Affiliations:** 1State Key Laboratory of Subtropical Silviculture, Zhejiang A&F University, Hangzhou 311300, China; 2Zhejiang Provincial Key Laboratory of Resources Protection and Innovation of Traditional Chinese Medicine, Zhejiang A&F University, Hangzhou 311300, China; 3College of Life Sciences, Yantai University, Yantai 264005, China

**Keywords:** pyruvate, acetoin, 2,3-butanediol, butanol, butyrate, L-alanine, metabolic engineering, microorganisms, sustainable platform

## Abstract

Pyruvate is a hub of various endogenous metabolic pathways, including glycolysis, TCA cycle, amino acid, and fatty acid biosynthesis. It has also been used as a precursor for pyruvate-derived compounds such as acetoin, 2,3-butanediol (2,3-BD), butanol, butyrate, and L-alanine biosynthesis. Pyruvate and derivatives are widely utilized in food, pharmaceuticals, pesticides, feed additives, and bioenergy industries. However, compounds such as pyruvate, acetoin, and butanol are often chemically synthesized from fossil feedstocks, resulting in declining fossil fuels and increasing environmental pollution. Metabolic engineering is a powerful tool for producing eco-friendly chemicals from renewable biomass resources through microbial fermentation. Here, we review and systematically summarize recent advances in the biosynthesis pathways, regulatory mechanisms, and metabolic engineering strategies for pyruvate and derivatives. Furthermore, the establishment of sustainable industrial synthesis platforms based on alternative substrates and new tools to produce these compounds is elaborated. Finally, we discuss the potential difficulties in the current metabolic engineering of pyruvate and derivatives and promising strategies for constructing efficient producers.

## 1. Introduction

Many chemicals and fuels are currently produced from fossil fuels, which will cause a worldwide decrease in fossil materials and many environmental problems. The emergence of metabolic engineering in the early 1990s provided ideas to solve the problems [1]. The establishment of the new discipline has significantly accelerated cell construction for chemical production. As a rapidly developing field, it uses genetic recombination technologies to change cell characteristics and combine with other technologies, such as biochemical engineering and microbial gene regulation, to construct new metabolic pathways for the synthesis of specific products [2]. To this end, a wide range of engineering strategies has been developed in various microorganisms, including the evolution of genome editing [3], tolerance engineering [4], rewiring of metabolic fluxes [5], and adaptive evolution [6].

Pyruvate is the hub of various endogenous metabolic pathways, which not only plays a crucial role in bioenergetic metabolism but also serves as a precursor for synthesizing a wide range of compounds. Compounds whose starting substance is sugar and whose target substance is pyruvate or are biosynthesized via pyruvate as an intermediate metabolite are called pyruvate-derived compounds such as acetoin, 2,3-BD, butanol, butyrate, and L-alanine. These derived chemicals are widely used in various fields. Acetoin and 2,3-BD have high potential applications as plasticizers, softeners, drug fumigants, and foods [7]. Butanol is used as a substitute for gasoline to alleviate the pressure of petroleum resources [8]. Butyrate mainly serves as animal feed additives, flavors, and pharmaceuticals while being a vital precursor for biofuel production [9]. L-alanine is a precursor of methyl glycine diacetate [10], which can be used as a novel synthetic green chelator.

Pyruvate is currently produced by chemical conversion or microbial fermentation to meet the fast-growing market demand. In terms of chemical conversion, pyruvate is mainly synthesized via dehydration and decarboxylation of tartaric acid [11]. In this process, pyruvate is distilled from the mixture of tartaric acid and potassium bisulfate at 220 °C. Then, the resulting crude acid is further distilled under a vacuum, which is unsustainable and depends on the extensive use of energy and dangerous solvents. In addition, using food crops like maize, potato, and cassava as raw materials for specific product fermentation will lead to the dilemma of competing with humans for food. The development of metabolic engineering can redesign microorganisms and engineer their metabolic ability to produce specific product production from renewable feedstocks. In this review, we outline the existing biosynthesis routes of pyruvate and derivatives and describe the metabolic engineering strategies in microorganisms to enhance productivity. We also propose a system-wide metabolic engineering strategy that combines systems biology and synthetic biology with molecular modification as a novel method for improving the production level, yield, and productivity of these compounds in the future. Detailed pyruvate and derived compounds production and engineering strategies are described in Table 1.

## 2. Metabolic Engineering Strategies to Enhance the Production of Pyruvate

### 2.1. Pyruvate Pathway and Its Regulation

Pyruvate is a central metabolite that has been reported in various microorganisms. The biosynthetic pathway of pyruvate is revealed in Figure 1. Pyruvate is synthesized from glucose through the glycolytic pathway. First, glucose is phosphorylated to generate glucose-6-phosphate via a phosphoenolpyruvate-dependent phosphotransferase system (PTS) or a coupled system of non-PTS and glucokinase (GLK). Then, glucose-6-phosphate is metabolized into pyruvate through glucose-6-phosphate isomerase (GPI), phosphofructokinase (PFK), fructose-1,6-bisphosphate aldolase (FBA), glyceraldehyde-3-phosphate dehydrogenase (GAPDH), phosphoglycerate kinase (PGK), phosphoglycerate mutase (PGM), enolase (ENO), and pyruvate kinase (PYK). Subsequently, pyruvate at the end of glycolysis is converted to various chemicals under different oxygen conditions. Under aerobic conditions, pyruvate is oxidatively decarboxylated to produce carbon dioxide (CO_2_) and acetyl-CoA that combines with oxaloacetate to produce citrate to enter the TCA cycle, a process that requires pyruvate dehydrogenase complex (PDHc), whereas this conversion is carried out by pyruvate formate-lyase (PFL) under anaerobic conditions. Excess pyruvate is catalyzed by α-acetolactate synthase (ALS) to α-acetolactate into the 2,3-BD pathway and catalyzed by pyruvate oxidase (POX) to acetate. Under anaerobic conditions, pyruvate is catalyzed to lactate by lactate dehydrogenase (LDH) in most bacteria, e.g., *Escherichia coli*, while pyruvate is catalyzed to acetaldehyde via pyruvate decarboxylase in yeasts. Pyruvate produces alanine via alanine transaminase (ALT) to enter the amino acid metabolism for cellular biosynthesis, which occurs under both aerobic and anaerobic conditions. In addition, pyruvate is catalyzed by pyruvate carboxylase (PYC) to oxaloacetate to balance intracellular acetyl-CoA, present in a few microorganisms, such as *Corynebacterium glutamicum* and *S. cerevisiae*. Oxaloacetate produces various organic acids through the TCA cycle, such as α-ketoglutarate, succinate, and malate.

The biochemical process of pyruvate production from glucose through the glycolytic pathway commonly follows the equation: glucose + 2 NAD^+^ + 2 Pi + 2 ADP → 2 pyruvate + 2 NADH + 2 ATP, indicating that the maximum theoretical yield of pyruvate from glucose is 0.966 g/g [57]. However, glucose is also used in microorganisms for other cellular substance formation. Therefore, according to the above equation, pyruvate can accumulate without other by-products, theoretically. Furthermore, the equation shows that NAD^+^ and ADP influence pyruvate formation. The NADH/NAD^+^ and ATP/ADP participate in enzymatic reactions as substrates or products and affect the metabolic network of the cell through the cyclic regeneration of cofactors. The glycolytic pathway synthesizes pyruvate to produce 2 mol ATP and 2 mol NADH. In addition, pyruvate enters the TCA cycle via acetyl-CoA to produce 1 mol ATP and 3 mol NADH under aerobic conditions. NADH enters the lactate and ethanol pathways for NAD^+^ regeneration via NADH-dependent dehydrogenase or produces ATP via oxidative phosphorylation (Figure 2). ATP satisfies cell growth and product synthesis while inhibiting the expression and enzymatic activity of GLK and PFK, which regulate glucose metabolism and pyruvate production efficiency.

### 2.2. Engineering to Expand the Glycolytic Flux for Pyruvate Production

Since pyruvate is located at the end of glycolysis as a direct product of glucose, its production depends on the glycolysis flux. Previous efforts to increase glycolysis flux focused on the overexpression of key enzymes. However, overexpressing critical enzymes of the glycolytic pathway is currently unable to increase flux due to the microorganism’s strict regulation of the carbon metabolism [58]. For example, overexpression of the enzymes PYK and PFK had a negligible effect on the increase of glycolytic flux [59]. Similarly, the dynamic regulation of GAPDH activity between 59% and 210% in *Lactococcus lactis* had no measurable effect on the glycolysis flux [60].

Subsequent uncoupling respiration in *E. coli* through mutations in the *atp* operon had been found to double the glycolytic flux [61]. Essentially, it is attributable to the fact that the relationship between ATP and glycolytic flux affects pyruvate accumulation. Accordingly, the researchers proposed that introducing *atp* operon mutations affect pyruvate formation. Introduction of the *atpA* mutation into *E. coli* produced more than 30 g/L pyruvate with a productivity of 1.2 g/L/h [12]. An effective alternative method to control intracellular ATP content is to target F_0_F_1_-ATP synthase. Inhibiting the activity of F_0_F_1_-ATP synthase in *Torulopsis glabrata* increased pyruvate productivity by 43.9% to 1.25 g/L/h [13]. Further development of an intracellular ATP content regulatory system is by regulating F_0_F_1_ATPase activity. One example is the expression of the *INH1* gene encoding an F_0_F_1_ATPase inhibitor from *S. cerevisiae* under a copper ion inducible promoter in the pyruvate producer *T. glabrata*; this approach increased pyruvate productivity to 1.69 g/L/h [14]. Another work on the combination of *atpFH* and *arcA* knockout is also worth mentioning. *E. coli* ALS929 knockdown of *atpFH* and *arcA* lacked oxidative phosphorylation but had hydrolytic F_1-_ATPase activity and showed an elevated glycolytic flux, producing 90 g/L pyruvate with a productivity of 2.1 g/L/h [15]. Higher intracellular pyruvate levels reduced the expression levels of key rate-limiting enzymes of glycolysis and were detrimental to increasing pyruvate productivity. Enhancing pyruvate production in *Candida glabrata* by vector engineering increases the rate of pyruvate transport to mitochondria [62].

### 2.3. Engineering to Increase the Carbon Flux for Pyruvate Accumulation

Since increasing glycolytic flux has limited pyruvate production, attention has been focused on other targets to direct more carbon flux to pyruvate. Therefore, it is necessary to prevent further pyruvate metabolism to increase its production. Undoubtedly, PDHc was selected as an early target to delete, because entering the TCA cycle via PDHc, which is composed of three subunits AceE, AceF, and Lpd, is the main catabolic pathway of pyruvate [63]. However, the growth of PDHc mutants usually requires the introduction of a secondary carbon source such as acetate or ethanol [64]. A wide variety of *E. coli* mutants deficient in components of the PDHc (*aceE*, *aceF*, *lpd* genes) growing with an acetate supplement led to a high pyruvate yield from glucose [16]. The best-performing strain *E. coli ΔaceF Δppc* generated 35 g/L pyruvate from glucose and acetate in 35 h at a yield of 0.78 g/g glucose [16]. Metabolic engineering approaches also use strategies that do not directly target PDHc, which often combine multiple features, including (1) preventing the formation of other by-products such as organic acids, ethanol, and amino acids [17], (2) disruption of the cyclic function of the TCA pathway, and (3) increasing glycolytic flux through uncoupled respiration. For example, *E. coli* Δ*pflB* Δ*poxB* Δ*ackA* Δ*ldhA* Δ*adhE* Δ*frdBC* Δ*sucA* Δ*atpFH* produced 52 g/L pyruvate in 43 h under anoxic conditions with glucose [18]. The major difference between the approach and those involving PDHc gene deletion is that maintaining PDHc activity avoids the requirement for acetate.

Knockouts usually represent a complete shutdown of the pathway and are less beneficial to the engineered bacteria than the regulated enzyme activity. Gene silencing is a promising metabolic engineering approach that has been proposed to replace gene deletion, which seems particularly suitable for targeting PDHc, because maintaining some activity of PDHc can support aerobic growth of the strain when glucose is the only carbon source. For instance, silencing the *aceE* gene, an essential gene encoding PDHc in *E. coli*, and combining it with the deletion of other genes produced 26 g/L pyruvate from glucose in 72 h [19]. The application of CRISPRi targeting the promoter of *aceE* or *pdhR* reduced *aceE* expression, resulting in a pyruvate yield of 0.36 g/g during glucose fermentation [20]. Gene silencing technologies are destined to become more widespread with the development of methods for fine-tuning the activity of target enzymes in metabolic engineering. Another approach to controlling the activity of key enzymes in a process is promoter engineering. A case in point was the modification of the promoter of the *aceE* gene to regulate PDHc expression in *E. coli*, which resulted in the production of 26.1 g/L pyruvate in 73 h [21]. An additional metabolic engineering strategy to redirect the carbon flux to the product was a point mutation of the *aceE* gene to reduce PDHc activity, thereby reducing carbon flow from pyruvate to acetyl-CoA, but not eliminating it [22]. The strategy to reduce PDHc activity usually needs to include the deletion of non-essential pathways.

### 2.4. Cofactor Engineering

Cofactor engineering is another method for increasing pyruvate production by changing the rate of NADH/NAD^+^. The conversion of glucose to pyruvate produces 2 mol NADH per mol glucose, and NADH is an inhibitor of the dihydrolipoamide dehydrogenase component of PDHc [65]. High levels of NADH markedly weaken the catabolism of pyruvate in microbes by inhibiting PDHc activity, while enhanced NADH oxidation to NAD^+^ increases the glycolysis flux [66].

Introducing an NADH or NAD^+^ regeneration system is a direct way to influence the NADH/NAD^+^ ratio. Intracellular NAD^+^ is easily regenerated in situ by expressing an NADH oxidase (NOX). The overexpression of *noxE* encoding exogenous NOX improved glucose consumption and glycolysis strength [67]. However, overexpression of NOX allows more carbon flow into the TCA cycle, resulting in no significant increase in pyruvate yield. Therefore, regulation of NOX expression in engineered strains is a more efficient strategy for pyruvate accumulation. Liu and Cao et al. introduced the *noxE* gene of *L. lactis* into *E. coli* MP-XB010 and produced 93 g/L pyruvate by regulating NOX expression through a temperature-controlled promoter [23]. The *noxE* gene was expressed at the beginning to promote the growth rate and biomass accumulation, followed by down-regulation to increase pyruvate synthesis rather than entering the TCA cycle [23]. Nevertheless, the issues of insufficient cofactor supply and cofactor imbalance are still significant bottlenecks for efficient biosynthesis. Dynamic regulation based on intracellular cofactor levels is a promising approach to solving this problem.

### 2.5. Engineering for Growth and Production Balance

Genetic optimization of metabolic networks to tailor natural microbial pathways is a conventional industrial microbiology strategy. However, the carbon flux balance between product synthesis and microbial cell growth is a persistent problem in previous industrial microbiology. The balance between production and growth requires precise manipulation of carbon fluxes, which can be achieved by regulating the expression of critical enzymes [68]. The vital enzymes for microbial growth require high levels of dissolved oxygen, which is detrimental to the accumulation of pyruvate. Hypoxia-inducible factor 1 (F1F-1) was designed to regulate multiple genes in the glycolytic pathway and the TCA cycle to enhance metabolic flux from glucose to pyruvate at low dissolved oxygen levels [24,69]. Further optimization of F1F-1 stability resulted in a titer of 53.1 g/L for pyruvate production in a 5 L bioreactor at 10% dissolved oxygen [24]. Transcription factors can also be used to regulate the expression of key enzymes in biosynthesis. Based on the pyruvate-responsive transcription factor *pdhR* from *E. coli*, a pyruvate-responsive bifunctional gene circuit was established in *Bacillus subtilis*, which could spontaneously respond to pyruvate concentration and dynamically regulate carbon fluxes in the *B. subtilis* [70]. 

In addition to using the metabolite-responsive transcription factor to regulate carbon flux balance dynamically, two-stage fermentation combined with static pathway engineering was used to balance cell growth and production. Pyruvate production was improved by a two-stage fermentation process using an *E. coli* strain that cut off the pyruvate consumption pathway [15]. The cell growth phase activated the TCA cycle by feeding glucose and acetate to obtain sufficient cell density, and the second phase disrupted the TCA cycle by restricting acetate feeding to accumulate products. However, the switch in microbial phenotype dependent on fermentation conditions may affect the entire metabolic network leading to unexpected effects on the strain. Therefore, it has been proposed that metabolic fluxes can spontaneously shift from growth-related pathways to production pathways during microbial production through a time-dependent trigger signal. To this end, a metabolic toggle switch was designed in an *E. coli* strain to shift metabolic flux from the endogenous pathway to the target synthetic pathway [25]. The researchers improved pyruvate production substantially by using a metabolic toggle switch to interrupt the TCA cycle and reduced pyruvate consumption by the endogenous pathway [25]. Alternatively, the trigger signal could be a quorum-sensing molecule that uses cell density to indicate cell growth. Quorum-sensing metabolic switches had been developed in *E. coli* and *S*. *cerevisiae* with applications for improved production of various compounds [71,72]. A similar trigger signal could be used to produce pyruvate and its derivatives in the future.

## 3. Metabolic Engineering for the Production of Pyruvate-Derived Compounds

### 3.1. Acetoin and 2,3-BD

There are two major pathways in the biosynthesis of acetoin and 2,3-BD (Figure 3). One pathway is that ALS condenses two pyruvate molecules to α-acetolactate, which converts to acetoin by α-acetolactate decarboxylase (ALDC). Another way is that α-acetolactate is spontaneously decarboxylated to diacetyl under aerobic conditions, which is then reduced to acetoin by diacetyl reductase (DAR). Finally, acetoin is catalyzed to 2,3-BD by 2,3-BD dehydrogenase (BDH). The biosynthesis of acetoin and 2,3-BD is vital for microbial physiology. The metabolic pathway of 2,3-BD may prevent intracellular acidification by changing the metabolism from acid production to forming neutral compounds [73]. Simultaneously, the reversible transformation between acetoin and 2,3-BD is associated with the cellular NADH/NAD^+^ ratio balance [74]. Furthermore, the synthesis of acetoin and 2,3-BD in cells can be considered an energy reserve strategy when glucose is depleted since acetoin is dissimilated to acetaldehyde and acetyl-CoA by the acetoin dehydrogenase enzyme system (AoDH ES) [75].

Many researchers have focused on enhancing acetoin and 2,3-BD production by designing the 2,3-BD pathway, such as overexpressing ALS or/and ALDC responsible for acetoin biosynthesis [26] or preventing the dissimilation of acetoin [76]. Based on this approach, a high ALS activity strain, *S. cerevisiae* YHI030, was constructed to promote the 2,3-BD pathway effectively, and the strain produced 81 g/L 2,3-BD [33]. Similarly, a 13.5-fold increase in the transcript level of ALsR in *B. subtilis* resulted in the highest increase of 17.1% for acetoin and 2,3-BD, and a high titer of 102.6 g/L for 2,3-BD was obtained by a three-stage oxygenation control strategy [34]. When energy sources are exhausted, the acetoin or 2,3-BD in cells will be reused as carbon sources, which is detrimental to the accumulation of acetoin or 2,3-BD. Destroying the AoDH ES encoded by *acoABCD* in *Klebsiella pneumoniae* blocked the re-utilization of acetoin and 2,3-BD [28]. The *acoABCD* mutant strain, *K. pneumoniae* Δ*bud* Δ*acoABCD,* produced 62.3 g/L acetoin after 57 h of fed-batch fermentation [28]. 

Cofactors are closely related to the 2,3-BD pathway. The production of acetoin and 2,3-BD could be improved by regulating the ratio of NADH/NAD^+^ [35,77]. Affecting the ratio of NADH/NAD^+^ by overexpression of endogenous or exogenous NOX received extensive attention. By introducing the *nox* gene from *Lactobacillus brevis* into *Serratia marcescens* H32, the NADH/NAD^+^ ratio in the cells was decreased 2.9-fold, and the acetoin production was increased by 33% [29]. When the *nox* encoding the water-forming NOX was selected from *L. lactis* and overexpressed in *C. glabrata*, NADH levels decreased by 62.6%, while the acetoin titer increased to 7.33 g/L [30]. In addition, acetoin in *K. pneumoniae* obtained similar results by introducing exogenous NOX [78]. According to recent research, a suitable substrate-decoupled system was designed to address the issue of NADH excess associated with the 2,3-BD pathway in *E. coli* without introducing an exogenous gene, and this strain produced 3 g/L of 2,3-BD [36].

The by-products such as acetate, ethanol, and lactate formed during the accumulation of acetoin or 2,3-BD in microorganisms reduce the availability of precursors and cofactors. Because the by-products are NADH-dependent products involved in the 2,3-BD pathway, it implies the possibility of reallocating carbon fluxes to target products by manipulating NADH levels. Expression of *yodc* encoding a new water-forming NOX in *B. subtilis* redistributed metabolic flux from its NADH-dependent by-product pool to the acetoin pathway [27]. The production of lactate and ethanol decreased by 70.1% and 75.0%, while the production of acetoin increased by 35.3%, reaching 56.7 g/L [27]. Other strategies to reduce by-products are by deleting the relevant genes. Deletion of *ldhA* encoding LDH, *budC* and *dhaD* encoding two BDH, and a *gcd* encoding glucose dehydrogenase in *Enterobacter aerogenes* reduced the conversion of acetoin to by-product and improved the acetoin production to 71.7 g/L [31]. Deletions of gene *pflB* encoding pyruvate formate lyase and gene *ldhA* in *K. pneumoniae* reduced the formation of ethanol and acetate by 88.8% and 99% but caused a 50% reduction in 2,3-BD productivity [37]. Similar results in another study had found that deletions of *ldhA* and *pflB* reduced the production of 2,3-BD [38]. The reason may be that the deletion of *pflB* prevents formate synthesis from pyruvate and leads to a significant reduction in the expression of genes involved in the formate hydrogen lyase [38]. Concurrently, because NAD^+^ regeneration is ineffective, the growth rate of the bacteria is reduced [77]. In addition, an *E. coli* strain was constructed by the promoter fine-tuning to balance cell growth and production for efficient production of 2,3-BD and no other by-products were formed from glucose [39].

### 3.2. Butanol and Butyrate

As revealed in Figure 4, the synthesis of butanol and butyrate with glucose as the carbon source involves two stages: the solvent-producing and acid-producing phases. First, pyruvate is generated through the glycolytic pathway, followed by acetyl-CoA. During the acid-producing phase, acetyl-CoA mainly produces acetate and butyrate. Acetate production is from acetyl-CoA conversion via phosphotransacetylase (PTA) and acetate kinase (ACK). The process of butyrate production is relatively complex. First, acetyl-CoA generates butyryl-CoA by the action of thiolase (THI), 3-hydroxybutyryl-CoA dehydrogenase (HBD), crotonase (CRT), and butyryl-CoA dehydrogenase (BCD). Then, butyryl-COA generates butyrate via phosphotransbutyrylase (PTB) and butyrate kinase (BUK). During the solvent-producing phase, acetate and butyrate are reabsorbed, then converted into acetyl-CoA and butyryl-CoA by CoA transferase (CTF). Next, acetyl-CoA generates ethanol in the action of acetaldehyde dehydrogenase (ALDH) and alcohol dehydrogenase (ADH), and butyryl-CoA generates butanol in the presence of butyraldehyde dehydrogenase (BLDH) and butanol dehydrogenase (BDH) or aldehyde/alcohol dehydrogenase (AdhE). Besides, acetone is produced from acetoacetyl-CoA via CTF and acetoacetate decarboxylase (AAD).

Butanol is generated in the solvent-producing phases. Enhancing the synthesis pathway and reducing by-products are fundamental to increasing butanol yield. On the one hand, overexpression of crucial enzymes of the synthesis pathway could increase butanol yield [41]. The cassette EC encoded by *thl*-*hbd*-*crt*-*bcd* was overexpressed in *Clostridium saccharoperbutylacetonicum*, resulting in a 13% increase in butanol production and a final titer of 17.4 g/L [42]. On the other hand, butanol accumulation was achieved by blocking the natural mixed acid pathway that competes with butanol synthesis for substrate and energy. Increasing the carbon flow to butanol synthesis by knocking out critical genes for succinate, lactate, acetate, and ethanol synthesis in *E. coli* significantly increased butanol production [79]. CRISPRi technology could also be used to repress the transcription of synthetic by-product genes, thereby reducing the synthesis of acetate, succinate, lactate, and ethanol [43]. The investigators engineered glyoxylate branches to further increase the carbon flow to the butanol synthesis pathway. Blocking the glyoxylate branching pathway by knocking down *aceA* in *E. coli* allowed more acetyl-CoA to be used for butanol synthesis [44]. Additionally, acetone is also a major by-product of butanol fermentation. The total butanol yield of *Clostridium acetobutylicum* increased from 57% to 70.8% after disrupting the *adc* gene-encoded AAD to eliminate acetone production [45].

However, increasing the concentration of butanol leads to severe product inhibition [80], because butanol reduces the surface tension of bacterial cell membranes to impair their function as cell barriers [81], further leading to ATP leakage and pH gradient disruption, thereby inhibiting cell growth. Therefore, the improvement of tolerance to butanol is necessary for butanol accumulation. Resistance to butanol stress can be achieved by altering membrane properties or controlling membrane-associated functions. Increasing outer membrane lipopolysaccharide content decreased cell surface hydrophobicity and prevented toxic hydrophobic compounds from entering the cytoplasm [82]. Butanol tolerance was increased in engineered strains with increased lipopolysaccharide content [83]. Mechanisms of butanol tolerance in *E. coli* were related to membrane-associated functions, including fatty acid synthesis and iron transport [84]. Moreover, the proteome and transcriptome analyses of *E. coli* under butanol stress showed that the gene expression and physiological responses of *E. coli* in response to butanol stress were similar to those in response to osmotic, oxidative, heat shock, and interference with respiration [85]. Overexpression of the heat shock protein gene in *E. coli* improved butanol tolerance of the strain [85].

Butyrate is generated through the acid-producing phase. Acetate, though, is produced as a by-product in this process, which reduces the butyrate yield and increases the cost of product recovery and purification. Additionally, end-product inhibition is one of the critical factors limiting the fermentation production of butyrate. Thus, it is also crucial to improve the strain’s tolerance to butyrate. The strains of *Clostridium* are commonly used for the metabolic engineering of butyrate. Earlier researchers reduced by-products by directly disrupting genes involved in the acetate synthesis pathway. The mutants of *Clostridium tyrobutyricum* with PTA or ACK deletion were constructed to allow more carbon flux into the butyrate pathway for increasing butyrate production, and mutant tolerance to butyrate was improved [86]. However, the trade-off was that the mutants suffered from a slower biomass growth rate, and the acetate production of the mutants did not decrease much [86]. Because acetate is reabsorbed into cells through conversion to acetyl-CoA during butyrate formation, it implies that metabolic pathways of acetate formation and conversion are indispensable for butyrate biosynthesis. Therefore, researchers sought to reduce acetate production by increasing the flux of acetyl-CoA to butyrate. For instance, overexpression of butyryl-CoA/acetate-CoA transferase and CRT in *C. tyrobutyricum* enhanced the flux of acetyl-CoA to butyrate and decreased acetate production, and the mutants exhibited higher specific growth rates and butyrate tolerance [46]. However, the lack of genetic engineering tools limited the prospect of metabolic engineering in *Clostridium* species. Therefore, some well-characterized microorganisms, such as *E. coli*, were engineered to produce butyrate. The metabolic engineering strategy used in *E. coli* consists of eliminating the acetate synthesis pathway and the major NADH-dependent reactions in metabolism and reconstructing a heterologous pathway targeting butyrate [47,48]. In such a modified butyrate pathway, acetyl-CoA is condensed to acetylacetyl-CoA, then converted to butyryl-CoA in three steps, and finally, butyryl-CoA hydrolysis to butyrate is catalyzed by *E. coli* thioesterase, unlike the *Clostridial* butyrate pathway. The natural thioesterase in *E. coli* contributes to the removal of coenzyme A from butyryl-CoA to produce butyrate directly [48]. However, strains utilizing the thioesterase pathway have lower butyrate productivity than those using the PTB/BUK pathway. This is because the additional energy produced by PTB/BUK favors a higher growth rate and cell density of the microorganism, which results in higher butyrate productivity [47].

### 3.3. L-Alanine

L-alanine is mainly synthesized for cellular biosynthesis and is converted using ALT in most microorganisms (Figure 5A). However, there is no practical application for a strain to produce L-alanine by using ALT because of the low L-alanine accumulation levels and the presence of the DL-alanine form. In addition to ALT, researchers had found that some organisms (e.g., *Bacillus sphaericus*, *Lysinibacillus sphaericus*, *Arthrobacter oxydans*, *Clostridium sp.* P2) can accumulate L-alanine in response to L-alanine dehydrogenase (ALD), which is converted to D-alanine by alanine racemase (ALR) [55,87,88,89,90], as displayed in Figure 5B. ALD-catalyzed pyruvate synthesis of L-alanine is the optimal pathway in microorganisms. However, many organisms have no ALD. Therefore, expression of exogenous ALD in strains is usually adopted for efficient accumulation of L-alanine. Such a production from pyruvate by NADH-linked ALD has been reported in various microorganisms.

Low levels of L-alanine were produced in two-stage fermentation by overexpression of the ALD gene *alaD* from *Bacillus sphaeroides* in *Zymomonas mobilis* CP4thi [51]. However, a significant drawback of the two-stage process is the large amount of carbon consumed for cell growth, whereas in the case of anaerobic fermentation, little of the carbon flux is recycled through the TCA for cell growth and most of it enters the designed product. It had been found that the natural D-lactate dehydrogenase promoter in *E. coli* could be used to regulate ALD, which was able to obtain high expression in anaerobic fermentation [88]. The obtained strain produced 114 g/L L-alanine in a mineral medium with more than 99.5% chiral purity [88]. Unfortunately, L-alanine accumulation inhibits the growth rate of cells and eventually leads to a decrease in L-alanine production. For the effective production of L-alanine, a thermoregulatory genetic switch was designed to dynamically control the expression of ALD from *Geobacillus stearothermophilus* in *E. coli* [52]. By changing the culture temperature, cell growth and efficient L-alanine accumulation could be well balanced. L-alanine titers reached 120.8 g/L when glucose is the only carbon source [52].

Another challenge in the fermentative production of L-alanine is the formation of by-products. Strains were created by deleting genes involved in competing for metabolic pathways to meet the challenge. After the *aceF* and *ldhA* genes were deleted in the *E. coli* mutant strain, 32 g/L of L-alanine was produced by a two-stage batch fermentation process [53]. Further, the pyruvate-formate lyase, phosphoenolpyruvate synthase, POX, LDH, and components of the PDHc were deleted in the *E. coli* mutants. The obtained strain produced 88 g/L L-alanine with a yield close to the theoretical maximum [54]. A similar strategy was also used for *C*. *glutamicum.* The genes for organic acid synthesis and ALR were deleted in *C. glutamicum*, resulting in carbon flux from organic acid to L-alanine and the production of 98 g/L L-alanine [55].

## 4. Alternative Sustainable Strategies for Pyruvate and Derivatives

### 4.1. Engineering of Alternative Substrates

Most of the metabolic engineering efforts for product synthesis have focused on using glucose as the substrate. Alternative substrates have not received the same attention, including other hexoses, pentoses, lignocellulose, glycerol, and CO_2_. Now, the utilization of alternative substrates was improved based on metabolic engineering by using microorganisms as a platform for sustainable industrial production (Figure 6).

Lignocellulose from forest biomass, agricultural residues, and marine algae is a potential alternative feedstock for the production of pyruvate and derivatives. The three main components of lignocellulosic biomass are cellulose, hemicellulose, and lignin [91]. Cellulose is the most abundant polymer and consists of glucose units. Hemicellulose consists of D-xylose, L-arabinose, D-mannose, D-galactose, acetyl groups, and glyoxylates, and lignin is a cross-linked phenolic polymer. D-xylose is the most abundant sugar in lignocellulose besides glucose. Therefore, efficient utilization of D-xylose is one of the prerequisites for the economic conversion of lignocellulose. However, most strains exhibit inefficient D-xylose fermentation. To this end, the expression level of D-xylose assimilation pathway enzymes was optimized by endogenous or heterologous metabolism [32,49]. Another difficulty is the inability to utilize all lignocellulose-derived sugars synergistically. The catabolic genes and manipulators that enable one to utilize pentose are often inhibited by carbon catabolites from more preferential carbon sources such as glucose, resulting in preferential utilization of glucose over pentose. Therefore, researchers strived to develop strains that metabolize glucose and pentose simultaneously, with typical strategies including supplementing the process with enzymes [92], slowing down glucose metabolism [93], and disrupting glucose-mediated carbon catabolite repression [94]. The previous reliance on modifications to the glucose transport pathway resulted in a significant decrease in glucose utilization by the engineered strains, thereby affecting strain growth. Current metabolic engineering strategies instead tend to preserve the glucose transport pathway. For instance, the combination of overexpression of PYK and increased L-arabinose catabolism achieved complete utilization of both glucose and arabinose [95]. Alternatively, pyruvate could be accumulated in a glucose–xylose mixture using a consortium of strains [96]. It was also found that inhibitors in lignocellulosic hydrolysates severely inhibit engineered bacteria’s cell growth and productivity [97,98]. Overexpression of Class I heat shock protein GroESL significantly increased its resistance to inhibitors derived from lignocellulosic hydrolysates [50]. Improving the tolerance of engineered bacteria to inhibitors and toxic substances is a future trend in the use of lignocellulose.

The industrial production of biodiesel is often accompanied by large amounts of crude glycerol, which can be used as a carbon source to produce essential platform chemicals [99]. During the fermentation of butyrate by *E. coli*, the butyrate yield increased with the addition of glycerol, which may be due to the improved protein folding by glycerol [100]. Further study of different mixed microbial cultures fermenting butyrate with glycerol as a carbon source resulted in a maximum yield of 11.14 g/L/d [101]. Biodiesel also could be used as an alternative carbon source in *Bacillus amyloliquefaciens* to produce 2,3-BD. A strain with high glycerol utilization was created by introducing the transcription regulator Alsr and the NADH/NAD^+^ regulation system into *B. amyloliquefaciens* [40]. Additionally, Zhou et al. investigated the effect of using glycerol as the sole carbon source on the fermentative production of L-alanine using *E. coli* [56]. After fermentation using a 5 L fermenter, 63.64 g/L L-alanine could be synthesized, and the conversion reached 0.63 g/g glycerol [56]. This study provides a significant reference value for the industrial production of L-alanine.

Most bio-industrial processes rely on the microbial carbohydrate metabolism to produce valuable chemicals. In this production scheme, the cost of the carbohydrate feedstock is a significant fraction of the total production cost. Using photosynthetic organisms provides an alternative production method that eliminates the cost of carbohydrate feedstock. In recent years, cyanobacteria have attracted significant attention because of their ability to use solar energy or CO_2_ as the sole energy source to produce biofuels and chemicals [102]. Researchers successfully used CO_2_ as a substrate to produce butanol from cyanobacteria. Photosynthetic butanol production was achieved by introducing a modified CoA-dependent butanol production pathway to the cyanobacteria *Synechococcus elongatus* PCC7942 [103,104]. Likewise, exogenous synthetic genes for 2,3-BD were introduced into cyanobacteria with appropriately controlled promoters and operons to direct the synthesis of 2,3-BD from CO_2_ [105]. Furthermore, a cyanobacterium capable of synthesizing 2,3-BD under light and dark conditions was constructed, and glucose or D-xylose was added to enhance the product yield and the industrial utilization of cyanobacteria [106]. As for the production of butyrate, the researchers successfully designed a model of the cyanobacterium *S. elongatus* PCC 7942 for direct photosynthesis to convert CO_2_ to butyrate with an optimal yield of 0.75 g/L [107], although the productivity of cyanobacteria is currently lower than that of other engineered bacteria. More research is necessary due to their independence and superiority in fixing carbon sources.

### 4.2. New Tools and Applications in Metabolic Engineering

Microbial electrosynthesis (MES) is an electrochemical process that relies on the interaction between electrodes and microorganisms to drive microbial metabolism for product synthesis [108,109]. As described in Figure 7A, the direction of electron flow in the MES is from the cathode to the electroactive microorganisms, which receive electrons and consume electricity. In MES systems, microbial conversion of CO_2_, CO and syngas help in efficient carbon capture and sustainable utilization. First, the bio-electrochemical production of butyrate using CO_2_ as a carbon source was achieved with a butyrate concentration of 20.2 mMC [110]. In a subsequent study, the supply of CO_2_ was controlled to establish high hydrogen partial pressure and promote butyrate production, resulting in the production of 87.5 mMC of butyrate [111]. A specifically mixed reactor microbiome was also developed, producing a mixture of isobutyric, n-butyrate, n-hexanoate, isobutanol, n-butanol, and n-hexanol using CO_2_ as the sole carbon source and the reducing power provided by the electrode [112]. Although the potential of MES shows the ability to drive relevant microbial industrial production, it is necessary to combine with genetic modification to optimize cellular metabolism for MES while controlling carbon and electron flow transfer pathways.

Conventional metabolic engineering is usually restricted to a limited number of genetic pathways. In contrast, genome-wide scale engineering promises a more comprehensive and rigorous design of microorganisms, which will undoubtedly involve more complex biological designs and larger datasets [113,114]. Adopting the traditional trial-and-error method leads to long development time and wasted substrate costs, which defies the concept of sustainability. For this purpose, machine learning (ML) is needed to provide outcome predictions and methodological recommendations for the development of metabolic engineering. ML is the learning of input datasets (e.g., genomics, transcriptomics, proteomics, metabolomics) through statistical algorithms, which in turn uses the generated empirical models to guide operations (Figure 7B), e.g., ‘learning’ the relationship between metabolites and enzymes from time series of protein and metabolite concentration data [115]. In the current research, ML with its capabilities in metabolic pathway reconstruction, metabolic flux optimization, and fermentation control has been increasingly used to solve problems in metabolic engineering and systems biology [116,117,118]. For example, ML was used to predict the behavior of syngas fermentation by *Clostridium carboxidivorans* for butyrate and butanol production [119]. To further maximize the sustainable effects of metabolic engineering, a range of factors are modeled at the beginning of the research phase, including the economic and environmental performance of the production of metabolically engineered products, even incorporating untested sustainability indicators at the design stage of the microorganism [120]. However, high-quality datasets should be generated for ML to provide more efficient predictability for sustainable metabolic engineering.

## 5. Conclusions and Prospects

This review aimed to present the progress of research on pyruvate and derived compounds based on metabolic engineering. Detailed pyruvate and derived compounds production and engineering strategies are described in Table 1. We found a long history of microbial production of these compounds. As shown in Table 1, *E. coli*, *C. glutamicum*, *C. glabrata, K. pneumoniae*, and *S. cerevisiae* are commonly used, producing strains, among which *C. glutamicum* is a food-grade microorganism with high safety and more suitable for industrial production. These microorganisms usually generate specific products by establishing synthetic metabolic pathways. Traditionally, microorganisms establish synthetic metabolic pathways mainly by gene overexpression or knockout to maximize production. However, improper gene expression or certain metabolic defects may cause problems such as cell growth inhibition, metabolic flux imbalance, and capacity reduction due to the inability to provide precise regulation. Therefore, metabolic engineering should shift from stepwise static regulation to dynamic regulation development, which helps maintain metabolic balance and cell growth. For example, switches in elements such as promoters can be induced upon exogenous signaling stimuli to regulate the expression of downstream genes, or dynamic regulation can be achieved by metabolite response elements such as promoters and transcription factors to regulate the expression levels of key enzymes of the pyruvate synthesis pathway, enabling cells to induce pyruvate spontaneously [70,121,122,123]. 

In addition, RNA-based engineering of *cis*-repressors has been successfully applied in metabolic engineering and synthetic biology to control specific proteins in a modifiable manner and alter metabolic fluxes to produce valuable chemicals [124,125]. The *cis*-repressors have the advantage of being portable and do not require an inducer due to the built-in set threshold for protein synthesis [125]. The complexity of regulation and metabolite crosstalk in metabolic pathways is also a major difficulty in the development of metabolic engineering. The recent advance of bacterial microcompartments has reduced the complexity by compartmentalizing intracellular enzymes using selective permeation of the protein shell [126,127]. The locus of bacterial microcompartments has been identified in 23 bacterial phyla [128], and will provide the basis for the next generation of metabolic engineering. The increase in the number of omics datasets, coupled with advances in ML, has also created tremendous opportunities for metabolic engineering. The increase in the number of omics datasets coupled with advances in ML have allowed insight into intracellular metabolic pathways, thus creating tremendous opportunities for scientists to explore cell-free biosynthesis systems in vitro [129]. Cell-free biosynthesis includes steps such as pathway design, enzyme mining, enzyme modification, multi-enzyme assembly, and pathway optimization, which will rely on metabolic pathway databases and applications of ML to design components and modules [130].

In addition, low-cost chemical methods allow the synthesis of high-purity pyruvate and derivatives [131], making them more economically attractive, which has led to limitations in the development of industrial fermentation. Therefore, reducing the cost of raw materials is an effective way to improve microbial metabolic engineering technology applications. For example, common agricultural waste lignocellulosic is used as a fermentation substrate [132]. However, other issues need to be considered to increase its industrial value: (1) developing more efficient technologies for saccharifying lignocellulose including the search for more efficient cellulose hydrolases; (2) developing pretreatment technologies to eliminate inhibitors and toxic substances from lignocellulose hydrolysates; (3) improving the tolerance of engineered bacteria to inhibitors and toxic substances to improve the efficiency of lignocellulose utilization; and (4) developing product recovery technologies to improve the efficiency of product recovery. In addition, some techniques are also under development for directly converting CO_2_ to acetoin, 2,3-BD, and butanol by some carbon-fixing microorganisms [106,133]. Although a suitable production route was successfully constructed, the cyanobacteria yield was too low to meet the needs of industrial production [134]. Therefore, we anticipate higher productivity if we develop methods to produce pyruvate and its derivatives from mixed substrates, combined with fermentation process optimization and downstream recovery operations.

## Figures and Tables

**Figure 1 molecules-28-01418-f001:**
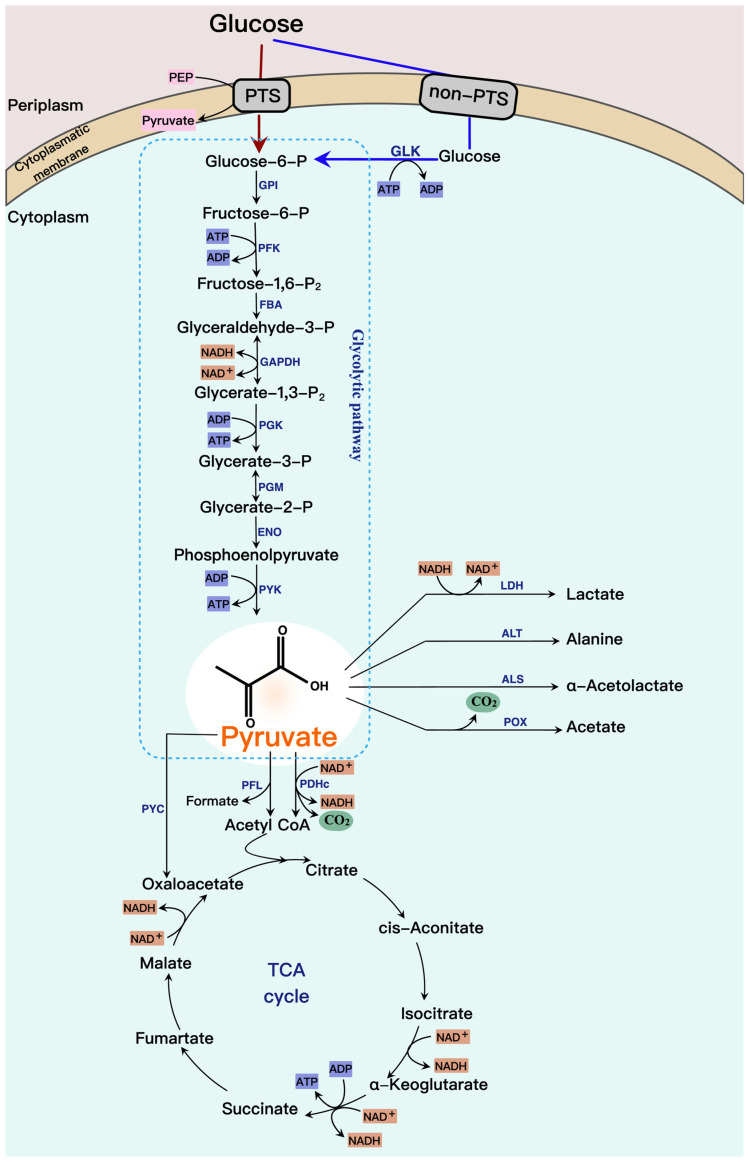
Synthesis and metabolism of pyruvate. The essential microbial metabolic pathways, enzymes, cofactors (e.g., NAD^+^ and NADH) and energy transfer compounds (e.g., ATP and ADP) are involved in pyruvate formation and consumption. Pyruvate is synthesized from glucose through the glycolytic pathway. Subsequently, pyruvate enters the TCA cycle via acetyl-CoA and converts to various chemicals such as lactate, alanine, α-acetolactate, and acetate.

**Figure 2 molecules-28-01418-f002:**
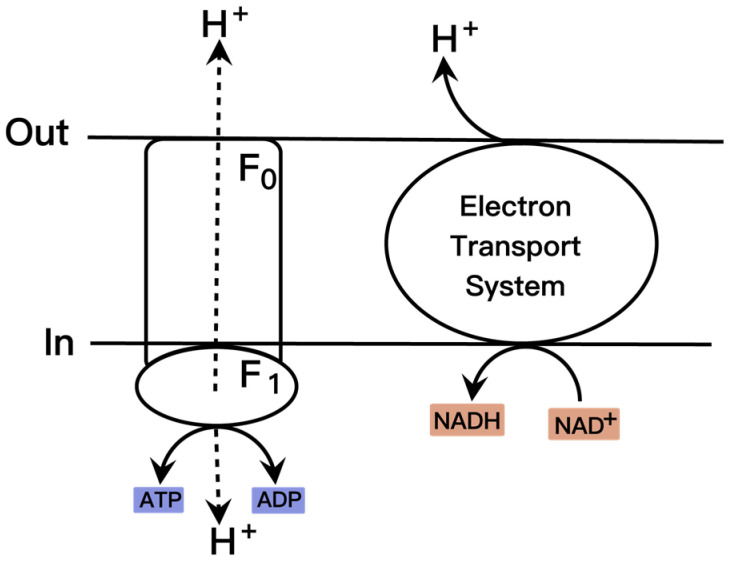
Oxidative phosphorylation of NADH in microorganisms.

**Figure 3 molecules-28-01418-f003:**
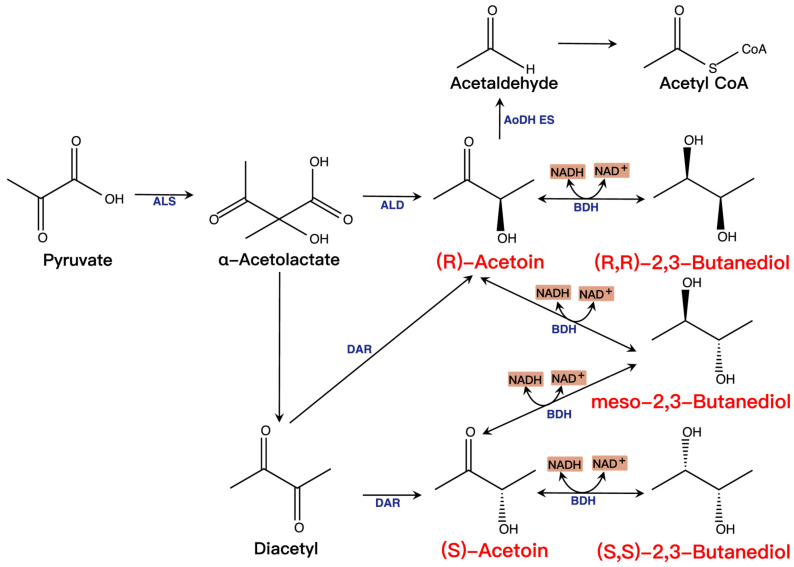
Synthesis and metabolism of acetoin and 2,3-BD. The essential microbial metabolic pathways, enzymes, and cofactors (e.g., NAD^+^ and NADH) are involved in acetoin and 2,3-BD formation and consumption.

**Figure 4 molecules-28-01418-f004:**
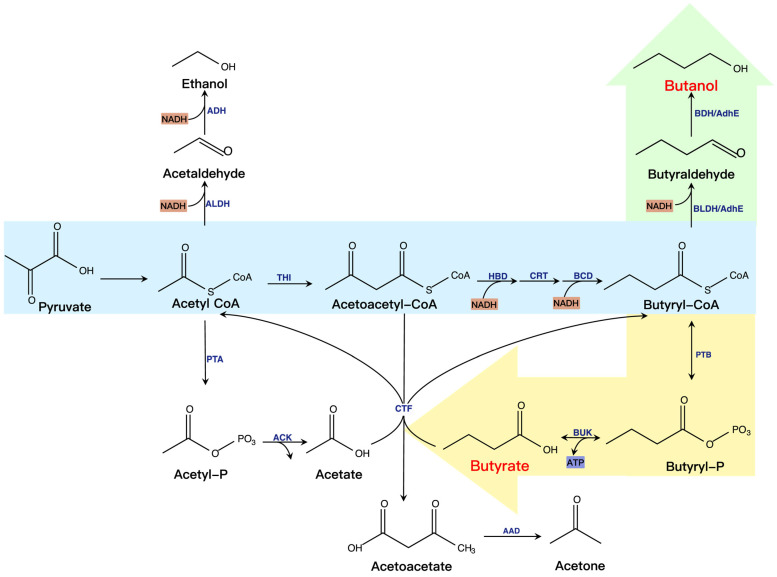
Synthesis of butanol and butyrate. The essential microbial metabolic pathways, enzymes, cofactors (e.g., NADH), and energy transfer compounds (e.g., ATP) are involved in butanol and butyrate formation. Green is the butanol synthesis pathway in the solvent-producing phases. Yellow is the butyrate synthesis pathway in the acid-producing phase.

**Figure 5 molecules-28-01418-f005:**
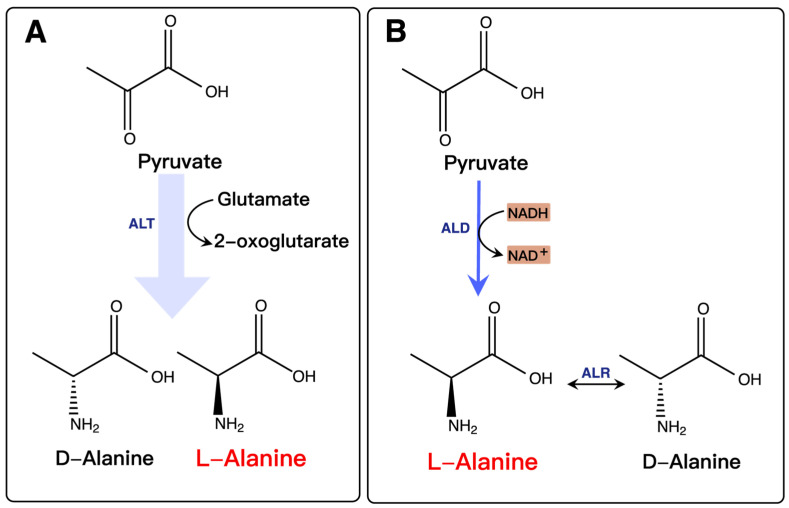
L-alanine synthesis by (**A**) alanine transaminase (ALT) and (**B**) L-alanine dehydrogenase (ALD).

**Figure 6 molecules-28-01418-f006:**
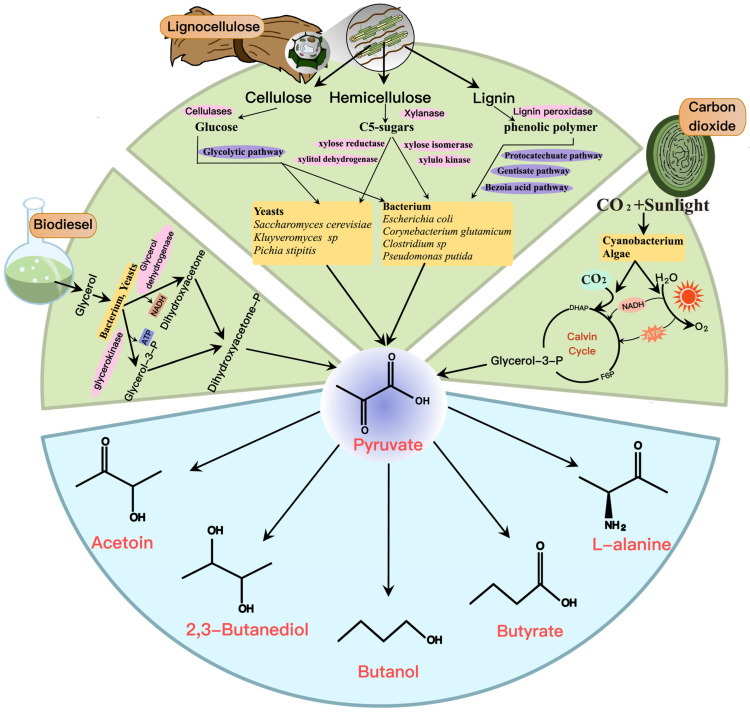
Alternative feedstocks for the production of pyruvate and derivatives, such as lignocellulose, biodiesel, and CO_2_. The yellow boxes are the microorganisms that have been implemented, the purple ovals are the important pathways involved in alternative feedstock utilization, and the pink ovals are the main enzymes involved in alternative feedstock utilization pathways.

**Figure 7 molecules-28-01418-f007:**
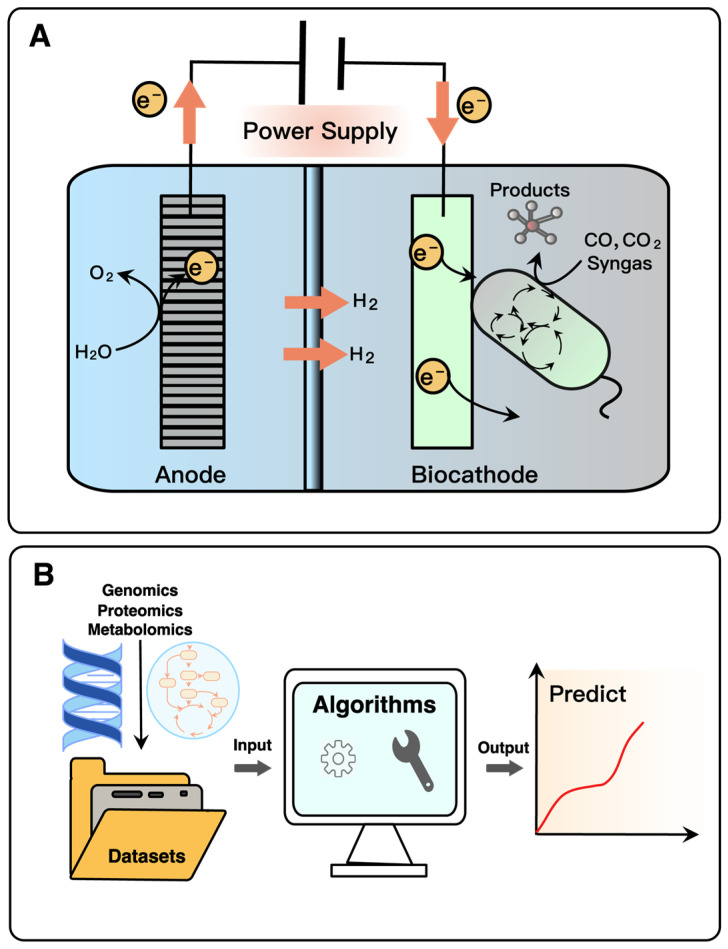
New tools in metabolic engineering: (**A**) microbial electrosynthesis (MES) and (**B**) machine learning (ML).

**Table 1 molecules-28-01418-t001:** Effects of different metabolic engineering strategies on products.

Strain	Engineering Strategy	Substrate	Culture Method	Titer (g/L)	Yield (g/g)	Productivity (g/L/h)	References
**Pyruvate**							
*E. coli* TBLA-1	*atpA* mutation	Glucose	Batch	30	0.64	1.2	[12]
*T.glabrata* N07	Reduced F_0_F_1_-ATPase activity	Glucose	Shake flask	49.8	0.52	1.25	[13]
*T. glabrata* INH1	Expression of *INH1* from *S. cerevisiae*	Glucose	Batch	67.4	ns	1.69	[14]
*E. coli* ALS929	Δ*aceEF*, Δ*pfl*, Δ*poxB*, Δ*pps*, Δ*ldhA,* Δ*atpFH,* Δ*arcA*	Glucose	Fed-batch	90	0.7	2.1	[15]
*E. coli* CGSC6162	Δ*aceF* Δ*ppc*	Glucose, acetate	Shake flask	35	0.78	1.2	[16]
*S. cerevisiae* Y2-15	Δ *PDC1*, Δ*PDC5*	Glucose	Shake flask	24.65	ns	0.26	[17]
*E. coli* W3110	Δ*pflB* Δ*poxB* Δ*ackA* Δ*ldhA* Δ*adhE* Δ*frdBC* Δ*sucA* Δ*atpFH*	Glucose	Fed-batch	52	0.76	ns	[18]
*E. coli* MG1655	↓*aceE*, ↓r *accA*, ↓*ppc*, ↓*gltA,* Δ *cra*	Glucose	Batch	26	ns	ns	[19]
*E. coli* MG1655	↓*aceE*, ↓*pdhR*	Glucose	Shake flask	11.28	0.33	ns	[20]
*E. coli* LAFCPCPt	tetracycline-regulated promoter regulates *aceE*,Δ*ackA-pta*, Δ*adhE*, Δ*cra*, Δ*ldhA,* Δ*pflB,* Δ*poxB*	Glucose	Batch	26.1	0.54	ns	[21]
*E. coli* ATCC 8739	Δ *ldhA*, Δ*poxB*, Δ*ppsA*, *aceE* point mutation	Glucose	Fed-batch	18.8	0.66	1.28	[22]
*E. coli* MP-XB010CN	Δ*ldhA*, Δ*pflB*, Δ*poxB*, Δ*ackA*	Glucose	Two-phase fermentation	93	0.71	2.02	[23]
*C* *. glabrata*	Engineering HIF1	Glucose	Batch	53.1	ns	ns	[24]
*E. coli* TA3052	Δ*dhA*, Δ*poxB*, Δ*pta*, Δ*adhE*, harboring *gltA*-OFF switch	Glucose	Shake flask	14.35	ns	ns	[25]
**Acetoin**							
*B. subtilis* PAR	↑*alsR*	Glucose	Shake flask	41.5	0.35	0.43	[26]
*B. subtilis* JNA 3-10 BMN	Δ*bdhA*, Δ*yodC*	Glucose	Batch	56.7	0.38	0.64	[27]
*K. Pneumoniae*	Δ*acoABCD*, Δ*budC*	Glucose	Fed-batch	62.3	0.29	1.09	[28]
*S. marcescens* H32	Introduction of the *L.brevis nox*	Glucose	Fed-batch	75.2	0.36	1.88	[29]
*C. glabrata*	Introduction of the *L. lactis nox*; ↑*PDC1*, ↑*GPD1,* Δ*adh*, Δ*ald*, Δ*bdh*	Glucose	Shake flask	7.33	ns	ns	[30]
*E. aerogenes* EJW-03	Δ*budC*, Δ*ldhA*, Δ*dhaD*, Δ*gcd*	Glucose	Fed-batch	71.1	0.32	2.87	[31]
*B.subtilis* BSL24	Introduction of the *Selenomonas ruminantium xsa*; introduction of the *Clostridium stercorarium xyn10B*	Xylose, xylan	Shake flask	15	0.3	0.11	[32]
**2,3-BD**							
*S. cerevisiae* YHI030	ΔPDC,↑*alsLpOp*, ↑*aldcLlOp*,	Glucose	Fed-batch	81	0.27	ns	[33]
*B. subtilis*	ALsR regulates the expression of ALS and ALDC; expression of *tdh* from *Clostridium beijerinckii*; Δ*ldhA*	Glucose	Three-stage fermentation	102.6	ns	0.93	[34]
*Klebsiella oxytoca* ME-UD-3	Δ*aldA*	Glucose	Fed-batch	130	0.48	1.63	[35]
*E. coli* BW25113	Δ*ldh*A, Δ*adhE,* Δ*frd,* ↑Ec-IlvBN, ↑Ec-GldA	Glucose	Shake flask	3	ns	ns	[36]
*K* *. oxytoca*	Δ*ldhA*, Δ*pflB*	Glucose	Fed-batch	113	0.45	2.1	[37]
*K. pneumoniae* KMK-05	Δ*wabG*, Δ*ldhA*, Δ*pflB*	Glucose	Shake flask	3.11	0.46	ns	[38]
*E. coli* W	Expression of *budA*, *budB* and *budC* from Enterobacteriaceae	Glucose	Fed-batch	68	0.4	4.5	[39]
*B. amyloliquefaciens* GAR	pMA5-*acr*-HapII-*dhaD*-P_bdhA_-*alsR*	glycerol	Fed-batch	102.3	0.44	1.16	[40]
**Butanol**							
*C. cellulovorans adh* E2	↑*fnr^CA^,* ↑*thlA^CA^,* ↑*hbd^CT^*	Cellulose	Shake flask	5.6	0.34	ns	[41]
*C. saccharoperbutylacetonicum* N1-4	↑*thl,* ↑*hbd,* ↑*crt,* ↑*bcd,* ↑*thl*, ↑*hbd*, ↑*crt*, ↑*bcd*, ↑ *adhE1,* ↑*adhE1^D485G^,* ↑*thl,* ↑*thlA1^V5A^,* ↑*thlA^V5A^*	Glucose	Bach	17.4	ns	ns	[42]
*E.coli* BW25113	↓*pta*, ↓*frdA*, ↓*dhA*, ↓*adhE*	Glucose	Bach	30	ns	ns	[43]
*E. coli* JCL299FT	Δ*ldhA*, Δ*adhE*, Δ*frdBC*,Δ*pta*, Δ*aceA*	Glucose	Shake flask	25.44	ns	ns	[44]
*C. acetobutylicum* EA2018	Δ*adc*	Glucose	Shake flask	12.2	0.203	ns	[45]
**Butyrate**							
*C. tyrobutyricum* ATCC 25755	↑*cat1*, ↑*crt*	Glucose	Fed-batch	46.8	13.22	0.83	[46]
*E.coli* LW393	Δ*ldhA*, Δ*frdABCD*, Δ*ackA*, Δ*adhE*; expression of *hbd*, *crt*, *ptb*, *buk*, from *C. acetobutylicum*; expression of *ter* from *T. denticola*	Glucose	Batch	33	0.37	0.89	[47]
*E. coli* BW *lacI^q^*	Expression of *phaA*, *phaB* from *Ralstonia eutropha*; *phaJ* from *Aeromonas caviae*; *ter* from *Treponema denticola*	Glucose	Fed-batch	12.34	0.313	0.23	[48]
*C. tyrobutyricum* Ct-pTBA	↑*xylT*, ↑*xylA*,↑*xylB*	glucose, xylose	Batch	42.6	0.36	0.56	[49]
*C. tyrobutyricum* ATCC 25755	↑*groESL*	corn/rice straw	Fed-batch	29.6/30.1	5.11/5.16	0.31/0.31	[50]
**L-alanine**							
*Z. mobilis* CP4thi	Expression of the *B. sphaericus alaD*	Glucose	Batch	8	0.16	ns	[51]
*E. coli* B0016-060BC	△*ldhA*, △*ackA-pta*, △*pflB*, △*adhE*, △*frdA*, △*dadX*::*cl*^ts^857-*p*R-*p*L-*alaD*-FRT	Glucose	Batch	120.8	5.03	4.18	[52]
*E. coli* AL887	Expression of the *B. sphaericus alaD*; △l*dhA*, △*aceF*	Glucose	Batch	32	0.63	ns	[53]
*E. coli* ALS929	Expression of the *B. sphaericus alaD;* △*pfl*, △*pps*, △*aceEF*, △*poxB*, △*ldhA*	Glucose	Fed-batch	88	4	1	[54]
*C. glutamicum*	Expression of the *L. sphaericus alaD*; ↑*gapA*, △*ldhA*, △*ppc*, △*alr*	Glucose	Fed-batch	98	ns	0.83	[55]
*E. coli* B0016-060BC	△*ldhA*, △*ackA-pta*, △*pflB*, △*adhE*, △*frdA*, △*dadX*::*cl*^ts^857-*p*R-*p*L-*alaD*-FRT	glycerol	Two-phase fermentation	63.64	0.63	1.91	[56]

ns, not specified; ↑, increased gene expression; ↓, decreased gene expression; △, gene deletion.

## Data Availability

Date sharing not applicable.

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
