# Peer review of "Metabolic Engineering of Microorganisms to Produce Pyruvate and Derived Compounds"

_molecules, 2023, doi:10.3390/molecules28031418_

Round 1

Reviewer 1 Report

1. Pyruvate metabolism – both aerobic and anaerobic metabolism should be mentioned. 

2. There are several issues that need the authors’ careful attention to improve the quality of this manuscript.

a. For this reviewer, the current title sounds weird. The authors may want to change it to “Metabolic engineering of E. coli to produce pyruvate and its derivatives”.

b. “Pyruvate derivatives” must be defined to provide readers a clear scope of this manuscript. Many (nearly any) chemicals can be produced from pyruvate.

c. In the abstract, regarding the statement “However, these compounds are chemically synthesized from fossil feedstocks, resulting in declining fossil fuels and increasing environmental pollution.”, what are these? At least one of the chemicals (i.e., alanine) is already commercially produced biologically. This issue also applies to the current title. “Conventional” may include both chemical and biological production for some chemicals.

d. In lines 124-126, it is not clear what exactly the sophisticated approach was. Did this study titrate the ATPase activity with Cu?

e. Line 127, collect the carbon flux? Maybe increasing?

f. Line 133, better than what?

g. Regarding paragraph 2.5, “growth-coupled production” is typically referred to as a strategy that cells are bound to produce a product to grow. The current paragraph seems to introduce multiple “optimal balancing” between production and growth, not coupling.

h. Line 256, acetate and lactate should be singular

i. Line 395, the subtitle needs to be specific to the section. The current one seems to cover the entire manuscript.

j. The most recent references need to be introduced. Ref 15 was published in 1932, nearly 90 years ago.

Reviewer 2 Report

The manuscript reviews recent literature in the microbial production of pyruvate and a few of the biochemical derivatives of pyruvate including acetoin, butanediol and alanine.  The approach is to be broad rather than deep.

11)    Line 76:  It is inaccurate and simplistic to state that acetyl-CoA “generate(s) organic acids such as a-ketoglutarate, succinate, and malate”.  Acetyl-CoA which enters the TCA cycle becomes CO2.  The organic acids are derived from oxaloacetate which is derived from pyruvate (if the microbe has pyruvate carboxylase) or from PEP (PEP carboxylase or PEP carboxykinase).   

22)    Line 80:  You note that pyruvate is catalyzed to oxaloacetate by pyruvate carboxylase.  This statement requires a modifier because the enzyme is not present in numerous pyruvate production strains, including E. coli.

33)    Figure 1:  The reaction indicated by pyruvate oxidase is incorrect in several ways.  Most importantly, the process generates electrons (it is an oxidation), and therefore it could never be accompanied by the conversion of NADH to NAD.  More specifically, most production strains (e.g., E. coli) contain EC 1.2.5.1 which mediates the reaction:  pyruvate + ubiquinone + H2O à acetate + CO2 + ubiquinol.  Finally, it is worth noting that the reaction constitutes a loss of CO2.  The reaction mediated by PDHC also is accompanied by the generation of CO2.

44)    Figure 1:  Glucose-6-P is conversion into Fructose-6-P not Flucose-6-P.  Similarly, Flucose-1,6-P should be rewritten as Fructose-1,6-P2 (not the subscript “2” following the P).

55)    Figure 1.  The reaction between malate and oxaloacetate is confusing because the two arrows point in opposite directions.  The typical conversion of malate to oxaloacetate is accompanied by the conversion of NAD to NADH, but the NADH should be at the top next to oxaloacetate.

66)    Line 272:  The sentence is unclear because the words “formate” and “pyruvate” are next to each other.  Perhaps the authors mean “…that the deletion of pflB prevents formate synthesis from pyruvate and leads to a significant reduction in the expression…”

77)    Figure 4:  Butyrate can be generated from butyryl-CoA via a native thioesterase without the need for a CoA transferase (see Kataoka et al. cited below).  This process has several advantages, and this pathway is not given much attention (merely lines 353-354).

88)    Figure 5B:  The arrow under ALR should be bidirectional.  The racemase is an equilibrium reaction.

99)    Figure 6:  Several errors appear in this figure.  Change “catechism” to “catechol”.  Change “celluloses” to “cellulases”.  Change “xylabases” to “xylanases”.  The molecular structure of butyrate and butanol are switched.  That is, the molecule appearing above the word “butanol” is butyrate, and the molecule appearing above the word “butyrate” is butanol.

110) The conclusion (section 5) does not contain any references.  However, several comments are made which clearly are associated with research results.  For example, the sentences “…switches in elements such as promoters can be induced…” (line 523), “…low-cost chemical methods allow the synthesis of high-purity pyruvate…” (line 520) and “although a suitable production route was successfully constructed…” all appear to refer to specific research, but no citation is provided.  Each of these and a few other statements should be cited.

There are a couple recent articles that should be included:

W. C. Moxley, M. A. Eiteman, Pyruvate production by Escherichia coli using pyruvate dehydrogenase variants, Applied and Environmental Microbiology, 87:e00487-21 (2021) https://doi.org/10.1128/AEM.00487-21

M. Ziegler, L. Hägele, T. Gäbele, R. Takors, CRISPRi enables fast growth followed by stable aerobic pyruvate formation in Escherichia coli without auxotrophy, Engineering in Life Sciences, 22(2): 70-84 (2022).  https://doi.org/10.1002/elsc.202100021

A.M. Erian, M. Gibish, S. Pflügl, Engineered E. coli W enables efficient 2,3-butanediol production from glucose and sugar beet molasses using defined minimal medium as economic basis. Microbial Cell Factories 17, 190 (2018). https://doi.org/10.1186/s12934-018-1038-0

N. Kataoka, A. S. Vangnai, T. Pongtharangkul, T. Yakushi, K. Matsushita, Butyrate production under aerobic growth conditions by engineered Escherichia coli, Journal of Bioscience and Bioengineering, 123:562-568 (2017). https://doi.org/10.1016/j.jbiosc.2016.12.008

The article by Zhu et al. (reference 39) showed 90 g/L pyruvate with a productivity of 2.1 g/Lh, and this example should be listed in Table 1.  Moreover, in this study the researchers knocked out the atpFH genes, which makes the work deserving for mention in the paragraph about this topic (lines 116-126.)

Reviewer 3 Report

Title: Metabolic Engineering of Pyruvate and Derived Compounds in 2 Microorganisms: from Conventional to Sustainable Production

The authors present in this work a review of the recent advances in the biosynthesis pathways, regulatory mechanisms, and metabolic engineering strategies for pyruvate and derivatives. This work could be of interest to people in the field and deserves being published. It is generally well written and clear. Some of the aspects that must be revised are the following.

Figure 4. Colors are hardly seen in my pdf version.

Line 365-6: “Such a production from pyruvate by NADH-linked ALD has been reported in various microorganisms.” Please, mention these organisms with the proper references.

Figure 6. I consider that this figure could be improved, e.g. mention in which organisms these pathways exist or have been implemented; mention main enzymes involved in the pathways.

Line 492. “ML has capabilities in metabolic pathway reconstruction, metabolic flux optimization and fermentation control”

DO you mean systems Biology?  I would be clearer as ML could be a general term and cause confusion. I would develop advances on System Biology a bit more in this part and I would change Figure 7B to indicate algorithms and different methods that must be integrated for ML (metabolomics, proteomics, etc.)-

Table 1 should be mentioned before, not in the conclusions section. I consider that data on table 1 could be handful for previous sections.

I would consider of interest to mention other approaches for metabolism control such:

Pandey N, Davison SA, Krishnamurthy M, Trettel DS, Lo CC, Starkenburg S, Wozniak KL, Kern TL, Reardon SD, Unkefer CJ, Hennelly SP, Dale T. Precise Genomic Riboregulator Control of Metabolic Flux in Microbial Systems. ACS Synth Biol. 2022 Oct 21;11(10):3216-3227. doi: 10.1021/acssynbio.1c00638. Epub 2022 Sep 21. PMID: 36130255; PMCID: PMC9594778.

Müller V. A synthetic bacterial microcompartment as production platform for pyruvate from formate and acetate. Proc Natl Acad Sci U S A. 2022 Mar 1;119(9):e2201330119. doi: 10.1073/pnas.2201330119. PMID: 35217629; PMCID: PMC8892506.

Kirst H, Ferlez BH, Lindner SN, Cotton CAR, Bar-Even A, Kerfeld CA. Toward a glycyl radical enzyme containing synthetic bacterial microcompartment to produce pyruvate from formate and acetate. Proc Natl Acad Sci U S A. 2022 Feb 22;119(8):e2116871119. doi: 10.1073/pnas.2116871119. PMID: 35193962; PMCID: PMC8872734.

References. I am a bit surprised that references from 2019 to 2022 are only 13 out of 136 while in the range 2014-2018 count on 47 out of 136. Why are so few references from the last years (2019-2022)?

There are recent reviews related to this topic:

Yuan W, Du Y, Yu K, Xu S, Liu M, Wang S, Yang Y, Zhang Y, Sun J. The Production of Pyruvate in Biological Technology: A Critical Review. Microorganisms. 2022 Dec 12;10(12):2454. doi: 10.3390/microorganisms10122454. PMID: 36557706; PMCID: PMC9783380.

Ku JT, Chen AY, Lan EI. Metabolic Engineering Design Strategies for Increasing Acetyl-CoA Flux. Metabolites. 2020 Apr 23;10(4):166. doi: 10.3390/metabo10040166. PMID: 32340392; PMCID: PMC7240943.

Tang S, Liao D, Li X, Lin Y, Han S, Zheng S. Cell-Free Biosynthesis System: Methodology and Perspective of in Vitro Efficient Platform for Pyruvate Biosynthesis and Transformation. ACS Synth Biol. 2021 Oct 15;10(10):2417-2433. doi: 10.1021/acssynbio.1c00252. Epub 2021 Sep 16. PMID: 34529398.

Sun L, Gong M, Lv X, Huang Z, Gu Y, Li J, Du G, Liu L. Current advance in biological production of short-chain organic acid. Appl Microbiol Biotechnol. 2020 Nov;104(21):9109-9124. doi: 10.1007/s00253-020-10917-0. Epub 2020 Sep 25. PMID: 32974742.

Andrews F, Faulkner M, Toogood HS, Scrutton NS. Combinatorial use of environmental stresses and genetic engineering to increase ethanol titres in cyanobacteria. Biotechnol Biofuels. 2021 Dec 17;14(1):240. doi: 10.1186/s13068-021-02091-w. PMID: 34920731; PMCID: PMC8684110.

Liu X, Xie H, Roussou S, Lindblad P. Current advances in engineering cyanobacteria and their applications for photosynthetic butanol production. Curr Opin Biotechnol. 2022 Feb;73:143-150. doi: 10.1016/j.copbio.2021.07.014. Epub 2021 Aug 16. PMID: 34411807.

Luo Z, Liu S, Du G, Xu S, Zhou J, Chen J. Enhanced pyruvate production in Candida glabrata by carrier engineering. Biotechnol Bioeng. 2018 Feb;115(2):473-482. doi: 10.1002/bit.26477. Epub 2017 Nov 6. PMID: 29044478.

Formal aspects.

Please, use italics when needed. Examples:

in situ, line 162

et al., line 167 and more
